# Fecal Zonulin as a Noninvasive Biomarker of Intestinal Permeability in Pediatric Patients with Inflammatory Bowel Diseases—Correlation with Disease Activity and Fecal Calprotectin

**DOI:** 10.3390/jcm10173905

**Published:** 2021-08-30

**Authors:** Edyta Szymanska, Aldona Wierzbicka, Maciej Dadalski, Jaroslaw Kierkus

**Affiliations:** 1Department of Gastroenterology, Hepatology, Feeding Disorders and Pediatrics, The Children’s Memorial Health Institute, 04-761 Warsaw, Poland; m.dadalski@ipczd.pl (M.D.); j.kierkus@ipczd.pl (J.K.); 2Department of Biochemistry and Experimental Medicine, The Children’s Memorial Health Institute, 04-761 Warsaw, Poland; a.wierzbicka@ipczd.pl

**Keywords:** zonulin, inflammatory bowel disease, biomarkers

## Abstract

Background: Recent data indicate that increased intestinal permeability plays a key role in the pathogenesis of inflammatory bowel diseases (IBD) and correlates with disease flare. Since zonulin related proteins (ZRP) are the proteins that increase permeability in the epithelial layer of the small intestine by reversibly modulating the intercellular tight junctions, they may serve as a new, noninvasive biomarker of disease activity. The aim of this study was to investigate fecal ZRP in pediatric IBD patients as well as its correlation with disease activity and fecal calprotectin (FCP). Methods: Ninety-four individuals: 47 Crohn’s disease (CD) patients, 41 ulcerative colitis (UC) patients, and 6 healthy controls were examined for fecal ZRP. Values were correlated to IBD type, disease activity for IBD patients, and FCP for all children included in the study. A stool specimen was collected the day before the visit to the hospital, then fecal ZRP and FCP were tested using the ELISA test. Non-parametric statistical tests were used for data analysis. Results: The level of fecal ZRP was higher among IBD patients compared to the control group (CG): medians for CD—113.3 (53.6–593.6) ng/mL; UC—103.6 (50.7–418.3) ng/mL; and CG—46.9 (31.8–123.0) ng/mL (*p* < 0.05). No difference in fecal ZRP concentration was observed between children with CD and those with UC (*p* = 0.55). A slight correlation between disease activity (PCDAI for CD and PUCAI for UC) and the fecal ZRP level was found for CD (*p* = 0.03/R = 0.33), but not UC (*p* = 0.62/R = 0.08), patients. A correlation between fecal ZRP and FCP was observed (R = 0.73, *p* = 0.00). Conclusions: Fecal ZRP levels are increased among those with IBD, are associated with CD activity, and strongly correlate with FCP. Further research into the role of intestinal permeability in IBD and the clinical usefulness of ZRP in IBD is warranted.

## 1. Introduction

Recent studies on the pathogenesis of inflammatory bowel disease (IBD)—Crohn’s disease (CD) and ulcerative colitis (UC)—indicate a significant role of increased intestinal epithelial permeability in disease development [1].

Zonulin is a 47-kDa human protein that increases permeability in the epithelial layer of the small intestine through reversibly modulating the intercellular tight junctions, whose proper functioning is crucial for maintaining physiologic processes in the intestine [2]. It is not exclusively the pre-haptoglobin 2 protein that is required, but a family of structurally and functionally related proteins called zonulin related proteins (ZRP); therefore, it is now recommended to indicate ZRP rather simply zonulin [3]. This term will be used further in the text.

Haptoglobin (HP) is a protein coding gene. Diseases associated with HP include Anhaptoglobinemia and Plasmodium Falciparum Malaria. The related pathways include the binding and uptake of ligands by scavenger receptors and the innate immune system, *serine-type endopeptidase activity,* and *hemoglobin binding*. An important paralog of this gene is HPR [4].

Increased serum/plasma ZRP levels have been found in celiac disease, type 1 and 2 diabetes, and in obesity-associated insulin resistance [5,6,7]. However, there is insufficient data on zonulin’s role in the development of intestinal inflammation for diseases such as IBD. Moreover, little is known about the correlation between serum and fecal ZRP or which one could be more useful in the diagnostics of IBD.

One pilot study has shown that serum ZRP is very sensitive for the assessment of intestinal permeability in IBD flares for both CD and UC, but no correlation between serum and fecal ZRP values was found [8]. On the other hand, the study conducted by a Czech team has demonstrated that both serum and fecal ZRP were elevated among CD but not UC, patients [9]. So far, there are no studies on ZRP in children with IBD.

Since the new European Crohn’s and Colitis Organization (ECCO) guidelines on therapeutics in CD emphasize the importance of noninvasive biomarkers in diagnostics (i.e., for the differentiation between organic and functional intestinal disorders), and in the monitoring of IBD activity [10], there is a great need for such markers. So far, the only validated and approved/recommended biomarker of intestinal inflammation is fecal calprotectin (FCP) [11]. Therefore, the invention of a new noninvasive biomarker would be of great benefit both from scientific (the discovery and validation of a new disease marker) and clinical/practical points of view.

Due to insufficient data and discrepancies concerning zonulin in IBD, we performed a study to evaluate fecal ZRP in pediatric IBD. Only the fecal form/concentration was used in our study since in children we search for the most noninvasive markers possible, preferably with no need for taking blood samples.

## 2. Patients and Methods

This is a single-center prospective observational cohort study of 94 children aged ≤ 18 years with IBD, either CD or UC, and 6 controls who were examined for fecal ZRP. Since we search for noninvasive markers of disease activity (such as FCP), especially in children, only fecal and not serum ZRP was measured. The control group included children admitted to an outpatient gastroenterological clinic by pediatricians. Among those children, no organic disease was diagnosed; most of them had functional abdominal pain or a rota/adenoviral infection. All participants collected a stool specimen the day before their visit to the hospital.

For IBD patients, disease activity was assessed using the pediatric Crohn’s disease (CD) activity index (PCDAI) or the pediatric ulcerative colitis (UC) activity index (PUCAI) [12].

Values were correlated to IBD type, disease activity for IBD patients (PCDAI, PUCAI), and FCP for all children included in the study.

### 2.1. Fecal Samples

The raw stool samples from the IBD and CG groups were frozen and stored at −80 °C after the sampling. The fecal ZRP was assessed using the competitive ELISA method (IDK^®^ Zonulin ELISA Kit, Immunodiagnostik AG, Germany). The ZRP results were given in ng/mL. Based on the manufacturer’s declaration, the intra-assay and inter-assay coefficients of variation were 3.4% and 13.3%, respectively [13]. The same method (the ELIZA test) was used to assess FCP levels, and values were given in μg/g.

According to the manufacturer, a median concentration of 61 ng/mL (±46 ng/mL) was estimated as a ‘normal’ value for fecal ZRP, and the ‘normal’ range for FCP was established to be <50 μg/g.

### 2.2. Statistical Analysis

The statistical analyses were performed using Statistica 12.0 (StatSoft, Krakow, Poland). Standard descriptive statistical analyses were performed, including frequency distributions for categorical data and calculations of the medians and interquartile ranges (IQRs) for continuous variables. The non-parametric Kruskal–Wallis (KW) test was used for the comparison of fecal ZRP and FCP concentrations in different subgroups of patients—CD, UC, control, and all-patient groups—while a Mann–Whitney tests were used for the comparison of continuous variables between two groups at a time—IBD vs. the control and CD vs. UC.

Multivariable regression was used to determine whether there was an important influence of age and to confirm that the regression assumptions were met.

Spearman’s rank correlation coefficient was used as a nonparametric measure of dependence between the variables examined. A *p*-value less than 0.05 was considered significant.

The primary outcome of the study was:− The assessment of ZRP, known for reflecting intestinal permeability, as a potential noninvasive marker of IBD and its activity

The secondary outcomes included:− The correlation between fecal ZRP and FCP− The correlation between fecal ZRP concentration and IBD clinical activity− Whether there is a difference in the fecal ZRP value between CD and UC

## 3. Results

The study included 94 individuals: 47 CD patients (50%), 41 UC patients (43.6%), and 6 controls (6.4%) at the mean age of 12.8 years (yrs) ± 3.9 yrs. The median fecal ZRP level of all participants was 109 ng/mL (31.8 ng/mL–593.6 ng/mL), whereas the median FCP concentration was 42 μg/g (2.0 μg/g–9801.0 μg/g). For patients with CD, the median PCDAI was 2.5 (0.0–52.5), and the median PUCAI for children with UC was 5.0 (0.0–40.0), which means that the majority of patients included in the study had mildly active disease at this time. (Table 1)

Fecal ZRP level was higher among IBD patients compared to control group (CG): median for CD—113.3 (53.6–593.6) ng/mL; UC—103.6 (50.7–418.3) ng/mL; CG—46.9 (31.8–123.0) ng/mL (*p* < 0.05) (Figure 1). No difference in fecal ZRP concentration was observed between children with CD and those with UC (*p* = 0.55). However, both fecal ZRP and FCP levels were higher among CD than UC patients.

A slight correlation between the disease activity (PCDAI for CD and PUCAI for UC) and the fecal ZRP level was found for CD (*p* = 0.03/R = 0.33), but not for UC (*p* = 0.62/R = 0.08) patients.

When analyzing the whole group of patients, a correlation between fecal ZRP and FCP was observed (R = 0.73, *p* = 0.00). However, a separate analysis of subgroups demonstrated that such a correlation was present only among IBD patients, not in the CG: CD—R = 0.8, *p* = 0.03; UC—R = 0.7, *p* = 0.02; CG—R = 0.5, *p* = 0.2. (Figure 2, Figure 3 and Figure 4).

Additionally, since the median age of the control group was younger than that of patients with IBD, multivariable regression was used to determine whether there was an important influence of age on fecal ZRP level, and it did not reveal such an influence.

## 4. Discussion

As far as we know, this is the first study assessing ZRP in pediatric IBD patients.

Our results have demonstrated higher fecal ZRP levels among IBD patients than among the control group, and there was a correlation between fecal ZRP and FCP.

Zonulin related proteins are the fecal proteins that reflect intestinal permeability, and their increased fecal levels are considered to be a marker of an impaired intestinal barrier, especially in the small intestine [12]. Increased serum/plasma ZRP concentrations have been found in different immunopathological diseases, such as food allergies, infections of the gastrointestinal tract, systemic autoimmune diseases, and inflammatory diseases of the intestine [13]. There are discrepancies in the correlation between fecal and serum ZRP levels [14]. Only a few works published so far describe zonulin use in IBD, and all of them include adult patients [6,7,15].

Caviglia et al. investigated the role of ZRP in patients with IBD and the correlation between its serum and fecal levels. Their study group included 118 IBD patients (86 CD and 32 UC patients) and 23 healthy controls (HCs). Authors have demonstrated that serum ZRP concentrations were higher among IBD patients compared to the HCs (34.5 [26.5–43.9] ng/mL vs. 8.6 [6.5–12.0] ng/mL, *p* < 0.001), and no correlation was observed between serum and fecal ZRPs (R = 0.15, *p* = 0.394) [3]. Our study, which assessed only fecal ZRP levels, has also shown higher ZRP concentrations among IBD patients than among the CG. Moreover, no difference in ZRP levels was observed between patients with CD and those with UC, both in our study and Caviglia’s study. However, since different forms/concentrations of ZRP were used in the studies, no objective conclusion can be drawn.

A Chech study from 2017 that examined 40 IBD patients and 40 healthy persons for fecal and serum ZRP concentrations has shown that both ZRP concentrations were elevated among patients with active CD but not for those with UC. A very interesting outcome from this study was the observation that smokers had high ZRP levels irrespective of IBD, which may point to the significant up-regulation of gut permeability in cigarette smokers [16]. In our study, fecal ZRP was elevated among both CD and UC patients; however, it was slightly higher in patients with CD (mean value for CD patients was 113.3 μg/g vs. 106.0 μg/g for UC patients). Moreover, we noticed a correlation between disease activity (PCDAI or PUCAI) and ZRP only among CD, and not UC, patients. These observations may be explained by the fact that zonulin is considered to be the best marker of increased permeability in the small intestine [17]. Since CD can extend to the whole gastrointestinal tract (including the small intestine), and the terminal ileum is the most common disease location among children, CD patients’ ZRP levels may be higher than those of UC patients’, which are restricted to the large intestine (with the exception of rare backwash colitis). Bowel healing, including that of the small intestine, may lead to a more significant decrease in fecal ZRP concentrations; thus, the correlation between CD clinical activity expressed by the PCDAI is better than that for the PUCAI for UC.

On the other hand, Wegh et al.—who investigated which markers were most relevant in assessing intestinal permeability among UC patients—have demonstrated that the serum, but not the fecal, ZRP level was elevated among those with active disease. Further, the serum showed a better correlation with other inflammatory markers, such as c-reactive protein (crp) [3]. Although in our study fecal ZRP concentrations were increased among UC patients, they did not correlate with the disease activity expressed by the PUCAI. Since we did not assess serum ZRP, we cannot state whether it may be a better marker of UC activity; this certainly needs further investigation.

Our study is the first one assessing ZRP in pediatric IBD, which is its great advantage. Due to the promising results with serum ZRP among IBD patients, we now consider conducting a similar study to assess ZRP serum concentrations in pediatric IBD patients and compare them to both fecal ZRP and FCP. The limitation of current our study is small sample groups, especially the control group. However, pediatric cohorts are always smaller than adult ones, and it is more difficult to enroll controls. We will enlarge our groups in further studies.

However, it is important to mention the limitations of the current commercial ZRP ELISA assays, which exercises caution in considering the measurement of serum ZRP as a marker of intestinal barrier integrity. The study by Ajamian M et al. [18] that investigated different ZRP assays demonstrated that all of them detected different proteins, none of which were ZRP. Therefore, there can be no value of circulating concentrations in assessing intestinal mucosal barrier dysfunction and permeability until the target proteins are indeed identified. [19] Commercial ELISA detection methodology may be improved with the development of specific and reliable monoclonal capture and detection antibodies for recombinant zonulin/prehaptoglobin-2 protein [19].

Taking into consideration all these results and the discrepancies between them, not enough evidence is available to draw any firm and objective conclusions on the role of ZRP as a potentially new, noninvasive biomarker of IBD activity. However, since it is a marker of increased intestinal permeability, ZRP is worth further research using properly designed studies.

## 5. Conclusions

In our study, fecal ZRP concentrations were higher among IBD patients than the CG, which may indicate that zonulin could serve as a new, noninvasive biomarker for IBD diagnosis. ZRP also associates with CD activity and strongly correlates with fecal calprotectin. Further research into the role of intestinal permeability in IBD and the clinical usefulness of fecal ZRP in IBD is warranted.

## Figures and Tables

**Figure 1 jcm-10-03905-f001:**
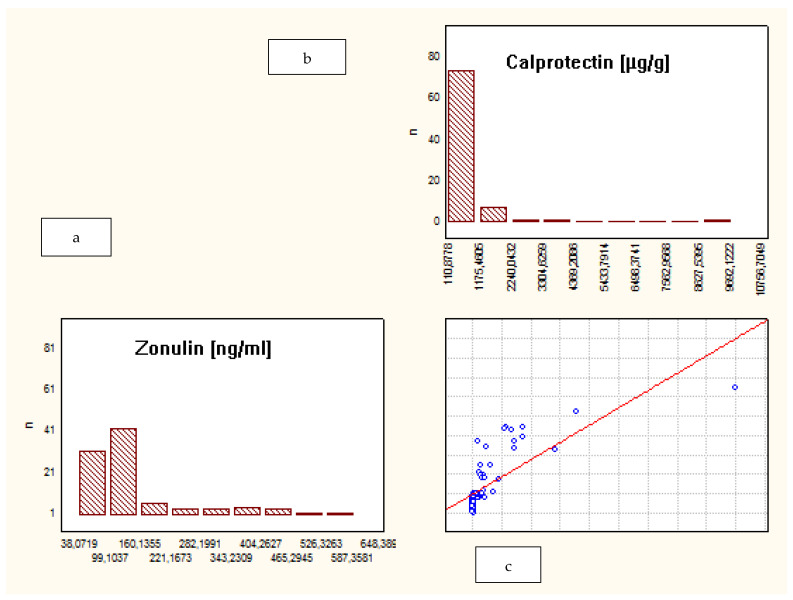
Correlation between fecal zonulin and calprotectin levels in all study participants. (**a**) Levels of fecal zonulin in all study participants (CD, UC, and control); (**b**) levels of fecal calprotectin in all study participants (CD, UC, and control); (**c**) the correlation between levels of fecal zonulin and calprotectin in all study participants.

**Figure 2 jcm-10-03905-f002:**
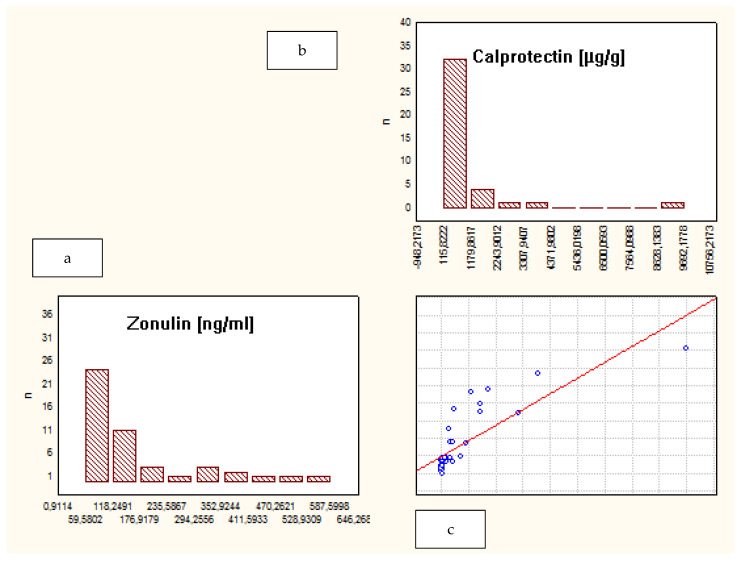
Correlation between fecal zonulin and calprotectin levels in patients with CD. (**a**) Levels of fecal zonulin in patients with CD; (**b**) levels of fecal calprotectin in patients with CD; (**c**) the correlation between levels of fecal zonulin and calprotectin in patients with CD.

**Figure 3 jcm-10-03905-f003:**
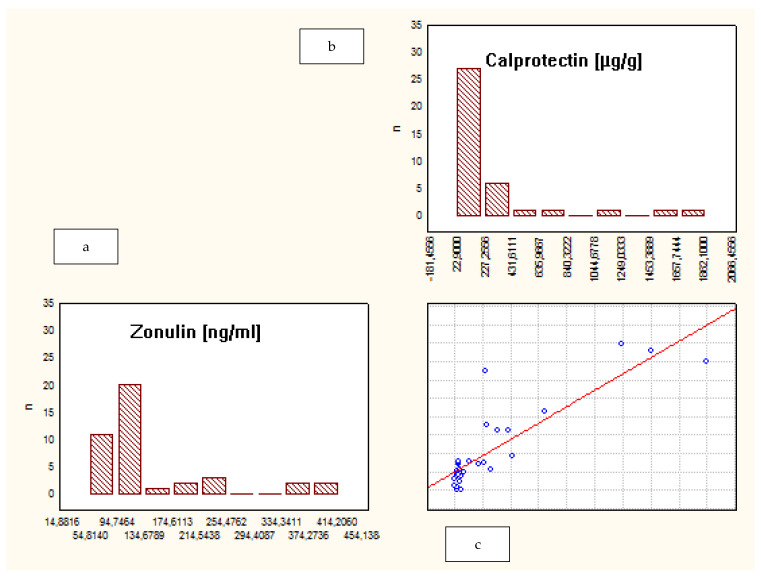
Correlation between fecal zonulin and calprotectin levels in patients with UC. (**a**) Levels of fecal zonulin in patients with UC; (**b**) levels of fecal calprotectin in patients with UC; (**c**) the correlation between levels of fecal zonulin and calprotectin in patients with UC.

**Figure 4 jcm-10-03905-f004:**
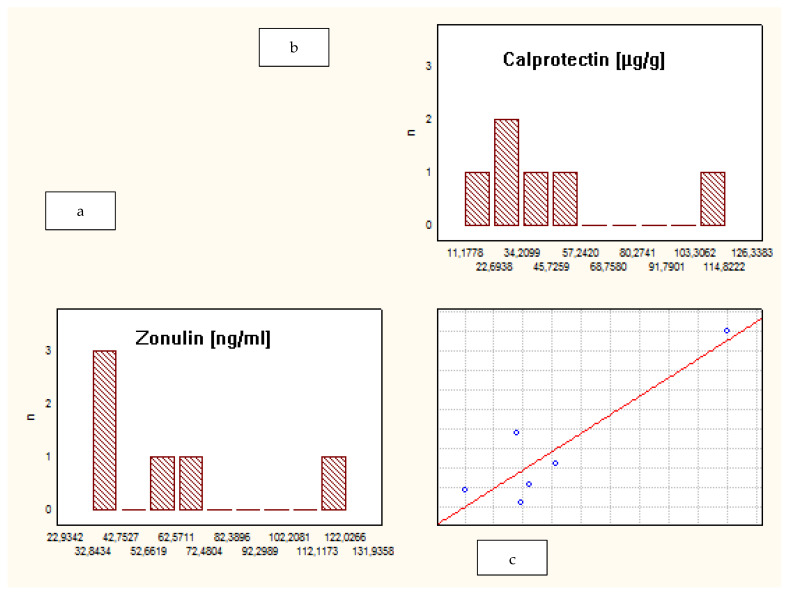
Correlation between fecal zonulin and calprotectin levels in the control group. (**a**) Levels of fecal zonulin in the control group; (**b**) levels of fecal calprotectin in the control group; (**c**) the correlation between levels of fecal zonulin and calprotectin in the control group.

**Table 1 jcm-10-03905-t001:** Characteristics of the study group.

Characteristic	Crohn’s Disease	Ulcerative Colitis	Control Group
Number of patients	47 (50%)	41 (43.6%)	6 (6.4%)
Median age, range (yrs)	14 (5.5–18.0)	14 (4.0–18.0)	8.5 (3.0–10.0)
Median FCP level, range (μg/g)	151.0 (71.0–9801.0)	39.0 (2.0–1883.0)	34.5 (10.0–116.0)
Median FZRP, range (ng/mL)	113.3 (53.6–593.6)	103.6 (50.7–418.3)	46.9 (31.8–123.0)
Median PCDAI, range	2.5 (0.0–52.5)	N/A	N/A
Median PUCAI, range	N/A	5.0 (0.0–40.0)	N/A

Abbreviations: Years, yrs; fecal zonulin related proteins, FZRP; fecal calprotectin, FCP; pediatric Crohn’s disease activity Index, PCDAI; pediatric ulcerative colitis index, PUCAI.

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
