# Peer review of "Fecal Zonulin as a Noninvasive Biomarker of Intestinal Permeability in Pediatric Patients with Inflammatory Bowel Diseases—Correlation with Disease Activity and Fecal Calprotectin"

_jcm, 2021, doi:10.3390/jcm10173905_

Round 1

Reviewer 1 Report

Szymanska and colleagues investigated the usefulness of the mesaurement of fecal zonulin in pediatric patients with IBD. The topic is clinically relevant and the concept of altered intestinal permability as cofactor for the onset and progression of IBD is getting recognized by the scientific community.

However, before possible acceptance of the manuscript, several major concerns need to be solved.

1) Recent evidences suggest that commercial assays for the measurement of serum/fecal zonulin detect a family of protein with structural and functional similarities to zonulin (Scheffler L et al 2018; Ajamian M et al 2019). As Prof. Fasano stated in Gut 2020, "zonulin is not exclusively pre-haptoglobin 2, rather is a family of structurally and functionally related proteins"; therefore, it is now recommended to indicate Zonulin related proteins (ZRP) rather simply zonulin (as updated also in the instructions of Zonulin IDK Immunodiagnostics). Please amend throughout the manuscript.

2) Introdcution. Lines 55-56. Really is FC considered a biomarker of increased intestinal permeability??

3) Unfortunately the cohort of control subjects is very small. Further, these subjects were recruited in a outpatient gastroenterological setting; they reported gastroenterological symptoms but without a diagnosis of IBD. More informations should be given for these patients, since zonulin has been found to be incresed also in patients with functional intestinal disorders or other organic intestinal diseases.

4) Materials and methdos. Please add references at lines 72 and 74.

5) The method used for zonulin measurement is a widely known commercially available ELISA. If the authors followed the manufacturers' instructions, there is no need to report all the protocol in detail. Please reduce paragraph length.

5) Statistical analysis. Was data normality checked? Which statistical test was used? KW test was used to analyze continuous variables; I suggest to use Mann-Whitney test for comparison of continuous variable between two groups, while to use KW for comparison of continuous variables between more than 2 groups.

6) Primary outcome. "Assessment of zonulin as biomarker of intesinal inflammation". Zonulin could be a biomarker of intestinal permeability associated to chronic inflammation in the setting of IBD....but is not a biomarker of intestinal inflammation. Many non-inflammatory conditions are characterized by incresed intestinal permeability and thus increased fecal/serum zonulin levels.

7) Secondary outcomes. Move lines 112-113 at the end of the manuscript in the appropriate section, as per JCM format.

8) Results. PCDAI and PUCAI values has been reported as mean+- SD, while in statistical analysis section it is reported that continuous variables were reported as median (IQR). Furthermore, in table 1 they are reported as median and range. Please be consistent.

9) Figures. All figures need to be improved in quality (low resolution). Furthermore, authors must add the Y and X axis labels for all panels. What the histograms refers to? Provide a description in the footnotes for all panels. In addition, looking at the scatter diagrams, it seems that the number of dots are not consistent with number of subjects included in the analysis. Were the data available for all subjects (please check all scatter diagrams in Figures 1, 2, 3 and 4)?

10) References. Check and modify the format of the references according to JCM guidelines. 

Author Response

please find the response attached

Reviewer 2 Report

Using patients fecal samples, the authors find that fecal zonulin (FZ) was increased along with fecal calprotectin (FCP) in IBD patients and proposing that FZ could be a noninvasive biomarkers of intestinal permeability.

They should shed more light on zonulin in the introduction (https://www.genecards.org/cgi-bin/carddisp.pl?gene=HP&keywords=Zonulin).

All the figures are of very poor quality; no X, Y axis labelling, poor pixel quality, which make the readers distracted. 

Author Response

please find the response attached

Reviewer 3 Report

The study by Szymanska et al. is the first to measure fecal calprotectin in children with IBD. It shows higher results in IBD relative to controls and may add to the literature. However, three major concerns need to be addressed first.

Major point no 1

The conclusion needs to be revised. The authors have not investigated the intestinal permeability (e.g., using a lactulose/mannitol test), but conclude “Fecal zonulin may serve as a new non-invasive biomarker of intestinal permeability in IBD”. This conclusion should be removed.

Using FZ as a surrogate for PCDAI makes little sense because PCDAI is easier to assess than FZ (one caveat: it requires a blood sample). The study could claim that “It may also be used to monitor disease’s clinical activity in CD patients” if endoscopic or histologic activity was assessed and FZ correlated strongly r > 0.6. Here, correlation with PCDAI is 0.33 (“slight” as the Authors point out themselves), and on this basis the conclusion is reached as above. It is important to note – someone in the future will read this paper and will think “FZ can be used to monitor CD activity” – patients and physicians may start to do the test even though what we know is that it correlates weakly with PCDAI and quite strongly with calprotectin. Please improve, this is an important point.

In my view, the conclusions could be: “FZ is increased in IBD, associates with CD activity and strongly correlates with fecal calprotectin. Further research into the role of intestinal permeability in IBD and clinical usefulness of FZ in IBD is warranted.” As it can be seen, the difference is important, because this conclusion does not claim that “FZ can be used to monitor disease clinical activity in CD”. Would you use FZ instead of calprotectin in your own patients?

Major point no 2

There is a concern that the kit used in this study does not actually measure zonulin. Please consider this article that comments on the Zonulin kit from Immunodiagnostik:

https://www.frontiersin.org/articles/10.3389/fendo.2018.00022/full?report=reader

While reading the abstract we see that the highest concentrations were 500 µg/g. This means that half of the mass of feces was zonulin! This is unbelievable. Moreover, IDK kit was shown not to correlate with measures of intestinal permeability:

https://pubmed.ncbi.nlm.nih.gov/33841181/

ELISA methods are tricky but this does not mean this research is not useful. It is important and timely and done in children. In my view, two actions should be taken to publish this study: (a) change the conclusions as above or even soften further (“may” “might” “seems”), (b) Add two sentences to the discussion: “Scheffler et al. showed that the IDK ELISA kit, which was employed in this study, detected mannose-associated serine proteases, including properdin [citation]. The protein levels that the kit has This strongly suggests that these proteins should also be further investigated in IBD, and findings regarding zonulin presented in our study (Precursor of Haptoglobin2) could benefit from further validation using another measurement method.” (or similar of course)

Major point no 3

Control group was much younger. What was done to compensate for confounding? Are regression assumptions met and multivariable regression can be used to check if there is important influence of age? Can stratification by age be done to verify higher FZ in IBD in a subgroup of young IBD patients vs controls?

Minor remarks:

L14 „disease flares”

L20 „disease activity for IBD patients” unclear – did you mean just “disease activity”?

L21 “in all” instead of “for all”?

L22 Lacks articles. Please proofread the manuscript.

L24 please use “µg” instead of “ug”

L56-57 Please provide a citation showing that fecal calprotectin is validated and approved/recommended biomarker of intestinal permeability.

L60 Did the study “evaluate the use of FZ in pediatric IBD” or just “evaluate FZ in pediatric IBD”? I think it is the latter. Evaluation of the use of FZ could be done by randomizing patients to receive FZ-informed care vs standard care, which was not done.

L61 “Only the fecal form/concentration was 61 used in our study since in children we search for most noninvasive markers possible, pref-62 erably with no need for blood samples taking.” This could be moved to the methods section.

L68 Were patients with diarrhea really used as controls? This makes it harder to prove the hypothesis presented in the manuscript (which is ultimately proven, however). This is just a suggestion or a point for discussion with reviewers (no changes in text needed), but have you tried to calculate AUC for the discrimination between IBD patients and children in whom IBD was suspected? Could FZ be used to diagnose IBD? I think no, but I prefer to make sure the Authors also thought about this issue.

L71 “In IBD patients”?

L98 “Statisctica” - „Statistica”?

L106-107 If such was the aim of the study then diagnostic value analyses should be performed, including ROC AUC analysis and/or sensitivity, specificity.

L117 Why FZ is given in µg/g in the abstract and in ng/g here? And in L124 it is also in micrograms. Which is true?

L118 9801.0 ug/g of calprotectin really? Impossible – it means 9,8 g/g.

L121 Max. PCDAI was 102. Is it possible if max PCDAI score is 100? Maybe I have overlooked something.  The same for PUCAI – why max PUCAI is 101? Max PUCAI should not be higher than 85. Maybe a patient with CD had PCDAI measured and was mistaken for a patients with UC, and then the value was assigned to PUCAI?

Fig. 1 quality is insufficient. No legends or captions can be read. (please ignore if irrelevant: Have you tried to log-transform one of the axes or both? Points could be better visible. You can do so even by adding two new columns in Excel – and each column is just a result of =LOG(variable))

L133 p=0.00 this is insufficient. What was the p value? If your software shows 0 maybe if you copy it to Excel it will show an exact value?

L155 citation 6,7,xii – this looks like a mixture of citation styles

L161 what does rs stand for? If Spearman’s rho then why is it not used elsewhere in the manuscript? (alternatives: “rho” or “ρ”) The levels in the study by Caviglia et al. are in nanograms. The Authors of this work present the results in micrograms. It is 3 orders of magnitude difference and is not discussed. Please review carefully what results you have, what other people have received and think it over, because your work is valuable but it is clear that the results were not thought over sufficiently yet.

L163 This is true that it looks like a consistent replication, very useful.

L192 “PCP” - FCP?

L197 “EBM” not necessary because it is not used in the rest of the manuscript – one may simply say “not enough evidence is available to draw…”

L202 “may indicate that zonulin could serve as a new non-invasive biomarker of intestinal permeability in IBD” – I suggest “may indicate that zonulin could serve as a new non-invasive biomarker of IBD activity” – as permeability was not measured.

L204-205 as in major remarks – the study does not show convincing evidence that FZ can be used to monitor CD. It suggests it could be useful and raises questions as to what it actually shows. Please keep in mind that the kit that you have used may actually not even measure FZ but something else! Thus, the conclusions need to be very carefully worded. This study is useful even without a strong conclusion.

Fig. 2 This really resembles a logarithm relationship. I suggest to try the transformations

This could be included in the discussion: https://pubmed.ncbi.nlm.nih.gov/27693318/ (not my work). You may also wish to check some information about zonulin inhibitors (developed for celiac disease), but I am not certain if this would add to the discussion.

The study uses a symptomatic control group and the Authors are right when they evade the use of the word “normal” with regard to the reference group.

At which time point were fecal samples collected relative to any preparation to colonoscopy?

Author Response

please find the response attached

Round 2

Reviewer 1 Report

The authors only partially improved the manuscript according to the comments received. For instance, the term zonulin has not been amended with Zonulin related proteins (ZRP) in all the manuscript. Furthemore, there are still important issues that need to be addressed. In particular regarding the previuous comment:

"All figures need to be improved in quality (low resolution). Furthermore, authors must add the Y and X axis labels for all panels. What the histograms refers to? Provide a description in the footnotes for all panels. In addition, looking at the scatter diagrams, it seems that the number of dots are not consistent with number of subjects included in the analysis. Were the data available for all subjects (please check all scatter diagrams in Figures 1, 2, 3 and 4)?"

Please provide a detailed description of the figures in the footnotes. More importantly, it does not seem that the dots depicted in the correlation figures are consistent with the number of patients analyzed. For instance, in figure 4 reporting the correlation between zonulin and calprotein in the control population (n = 6), I can see more than 6 dots....

Author Response

please find the response attached

Reviewer 2 Report

Reviewer: All the figures are of very poor quality; no X, Y axis labelling, poor pixel quality, which make the readers distracted.

>>Authors: The figures are made in Statistica program therefore little is possible to provide any changes to them

Reviewer: I am not convinced that the figure quality can not be changed. The author can try with another program (i.e. Graph Pad, etc; ). Even they did not label it yet; labelling can be done with Power point in any kind of file (PDF, Tif etc).

Author Response

please find the response attached

Reviewer 3 Report

Despite the revision and a significant correction of the conclusions, some important issues persist. 

Major issue A

I will treat the first issue at some length, because in my view this point is really important. It boils down to including one or two sentences in the limitations of the study. It is all I asked for, but it was not done.

The Authors seem reluctant to acknowledge that IDK FZ has limitations, for which I have asked, and for which I indicated evidence in the published literature. I have only asked to inform the reader that interpretation should be cautious. I have not asked to reject the study or repeat it, but to say the truth. If this is not done, then the reader will be uninformed. Inclusion of information about this would be sufficient. It is clear that many people use IDK FZ. This is not a strong argument. The problem is not to wish to say that IDK FZ should be interpreted very, very cautiously. If the Authors know that there is a potentially serious issue (they know because I have indicated articles that seem to prove this), but do not wish to inform the reader about this, the study should not be published.

Therefore, major point no. 2 from the last revision has not been answered satisfactorily. I indicated two articles, of which one shows that IDK FZ does not measure zonulin and another, which shows that it does not measure intestinal permeability. The Authors replied that the kit is "validated". Please cite an academic article validating IDK FZ test (where someone used IDK FZ and used another, reliable method of FZ assessment, and showed very good correlation). I have found such work and it shows that the kit is almost useless, please cite another study that shows that IDK FZ really measures FZ, and not something else. If you are unable to show IDK FZ is validated, please include a statement, in the limitations (discussion section), indicating that there have been some doubts about IDK FZ and that this work could benefit from further validation (I have suggested how this can be written in the previous revision). This does not invalidate the current study. It is something that a reader needs to know. If I used an ELISA kit "ferritin" and produced an article about "ferritin" and then I learned that the ELISA kit actually measured GM-CSF, would it be OK to publish article about "ferritin" without mentioning that according to some it is GM-CSF and not ferritin that is measured by this kit? The kit may not measure FZ and we cannot do anything about it now that the study is done. But the reader should be informed to remain very cautious in study interpretation.

Major issue B

The quality of figures is very poor. It is almost impossible to read the text, but if the effort is made to try to read, then it seems that the text is not in English. There are no labels for axis ticks (what are the values shown, units?). In my view, the quality of figures is unacceptable and they must be replaced at this stage if publication is to be considered.

The Authors stated that they used Statistica and "little can they do to change them". Normally, a reviewer would answer she/he has not seen such bad figures in any respectable journal and that the work should be rejected and that "little can she/he do to change this". I do not ask for a minor improvement, where saying "it will not be done" is fine. These figures cannot be read and I do not know what is on which axis and what is the level measured, etc. Every reader will have trouble trying to interpret them and will see this is a serious problem. This should be improved. If Statistica is not good enough, maybe other software should be used?

A FAQ on Statistica indicates a solution: "When saving the graph as a jpeg, there is optional setting to define resolution (dpi) of final image." ( https://community.tibco.com/wiki/file-types-tibco-data-science-statistica )
So it is possible to have better resolution (like 300 DPI), I hope this works and helps the Authors. Please improve the figures. It should be clear what is on the figure, what do the axes show, what are the values (and what are the values) - the figures require an important correction. You may also wish to try to plot log-transformed variables (not necessary, but possibly a useful hint to improve the work even further).

Other remarks

L45-L46 "Systeme" - should it not read "System"?

L104 - The study does not measure intestinal permeability. If the primary outcome was to assess FZ as a measure of intestinal permeability in children with IBD, then lactulose/mannitol test should be performed and agreement between FZ and L/M be assessed. Therefore, the main aim of the study was simply "Assessment of zonulin, known for reflecting intestinal permeability, as a potential noninvasive marker of IBD and its activity" or similar, if you agree.

L185 - why is there citation "3" and not in the Roman numerals as elsewhere? This should be corrected throughout. Or maybe it is not a citation?

L93 - I highlighted "Statisctica" because there is error. It should be "Statistica" and not "Stati-s-c-t-i-ca" as currently.

When asked about ROC AUC analysis the Authors replied that the figures should be consulted: "See the figures". I have went through all the figures and there is not one single ROC curve. There is no mention of sensitivity or specificity in the manuscript. I think this comment was not addressed correctly. If diagnostic value is measured, then at least sensitivity and specificity should be provided, and ROC analysis are typical in such cases also. I would expect that at least sensitivity and specificity for diagnosing IBD (distinguishing between IBD and healthy controls) is provided. ROC is not required, but can be added.

It is interesting that the highest FZ, at 9,8 mg/g was in a patient with very severe CD, thank you for your reply.

I understand there was a typo regarding PCDAI. But please also think about PUCAI. Your maximum measured PUCAI was 101 (L 117 and Table 1 in the bottom). PUCAI cannot be higher than 85, if I remember correctly.

"the term “normal” group was not uded" - exactly! I think this is done very well, this is why I wrote "The study uses a symptomatic control group and the Authors are right when they evade the use of the word “normal” with regard to the reference group."

I will recommend a major revision and I hope that the changes listed above can be introduced and that the work will be published. However, if these issues persist (concerns about IDK FZ not mentioned in one or two sentences with adequate citation/s, poor figure quality and resolution, lack of sensitivity/specificity for IBD diagnosis), I will recommend a rejection, if given such opportunity.

Author Response

please find the response attached
